# Effect of Dietary Incorporation of Linseed Alone or Together with Tomato-Red Pepper Mix on Laying Hens’ Egg Yolk Fatty Acids Profile and Health Lipid Indexes

**DOI:** 10.3390/nu11040813

**Published:** 2019-04-10

**Authors:** Besma Omri, Raja Chalghoumi, Luana Izzo, Alberto Ritieni, Massimo Lucarini, Alessandra Durazzo, Hédi Abdouli, Antonello Santini

**Affiliations:** 1Laboratory of Improvement & Integrated Development of Animal Productivity & Food Resources, Higher School of Agriculture of Mateur, University of Carthage, Tabarka Road, Mateur, Bizerte 7030, Tunisia; omribesma1@gmail.com (B.O.); chalghoumi.r@hotmail.com (R.C.); abdoulihedi@gmail.com (H.A.); 2National Agronomy Institute, University of Carthage, Avenue de la République, P.O. Box 77, Amilcar, Tunis 1054, Tunisia; 3Department of Pharmacy, University of Napoli Federico II, Via D. Montesano 49, 80131 Napoli, Italy; luana.izzo@unina.it (L.I.); alberto.ritieni@unina.it (A.R.); 4CREA—Research Centre for Food and Nutrition, Via Ardeatina 546, 00178 Rome, Italy; massimo.lucarini@crea.gov.it (M.L.); alessandra.durazzo@crea.gov.it (A.D.)

**Keywords:** fatty acids, yolk, pepper, tomato, thrombogenic, atherogenic

## Abstract

This study evaluated the effect of linseed incorporation in laying hens’ feed (alone or along with a tomato-red pepper mix) on laying hens’ egg yolk fatty acids profile, as well as on their atherogenic (IA) and thrombogenic (IT) health lipid indexes, and the ratio between the hypocholesterolemic and hypercholesterolemic fatty acids (HH). Sixty 27 weeks-old Novogen White laying hens were divided into three groups and given 100 g/hen/day of a standard diet (Control, C) containing 4.5% of ground linseed (Linseed diet, L), containing 1% of dried tomato paste and 1% sweet red pepper (Lineseeds-Tomato-Pepper, LTP). The linseed dietary inclusion significantly reduced the egg yolk content of palmitic acid from 25.41% (C) to 23.43% (L) and that of stearic acid from 14.75% (C) to 12.52% (L). Feeding 4.5% ground linseed did not affect the egg yolk content of α-Linolenic acid but significantly increased the egg yolk concentration of eicosapentaenoic acid (EPA) from 0.011% (C) to 0.047% (L) and that of docosahexaenoic acid (DHA) from 1.94% (C) to 2.73% (L). The IA and the HH were not affected (*p* > 0.05) by the dietary addition of linseed, whereas the IT decreased (*p* < 0.05) from 1.16 (C) to 0.86 (L). Adding tomato-sweet red pepper mix to the linseed-supplemented feed did not affect the measured parameters as compared to the linseed dietary treatment.

## 1. Introduction

Hen’s eggs that provide nutrients have been used as a food by human beings since antiquity. Eggs have been well accepted as a safe and nutritious food for all ages. Eggs are readily available and are a balanced and nutrient rich inexpensive food; they have a great culinary versatility, and low economic cost. This make eggs widely accessible to most of the population [1]. In addition, they are accepted worldwide and are not subject to major cultural or religious prohibitions [2].

Hen’s eggs are also considered a functional food, since they are a source of high-quality proteins, vitamins, minerals and lipids, such as phospholipids and polyunsaturated fatty acids (PUFA) [3,4,5,6,7]. Beside their nutritional value, the antimicrobial, immunomodulator, antioxidant, anti-cancer or anti-hypertensive effects of some active components of eggs were also reported [8,9,10,11]. For instance, Nimalaratne and Wu. [11] reported the presence of several compounds in both egg white and yolk exhibiting antioxidant and nutraceuticals properties [12,13,14,15,16,17,18,19,20]. However, eggs from hens fed a standard diet contain substantial amounts of cholesterol (ranging from 183 mg/egg [21] to 386 mg/egg [22,23]) and of saturated fatty acids (around 150 mg/egg) [24,25] exerting a hypercholesterolaemic effect.

Nowadays, the worldwide interest in enriched eggs is growing tremendously. The enrichment or the biofortification consists of adding physiologically active components that improve the health status and well-being of the consumers. Omega-3 fatty acids and carotenoids are among the many bioactive compounds used for egg biofortification [26,27], together with byproducts of the agro-system chain, in an environmentally friendly way [28,29,30,31,32].

Previous studies concluded that the fatty acid composition of eggs is dependent on the fatty acid composition of the feed of the laying hen [33,34,35,36,37,38,39]. Omega-3 fatty acids are one of the most sought after components in the functional food sector. Ramirez et al. [39] reported that egg lipids are efficiently absorbed in the body, increasing the bioavailability of DHA and levels of high-density lipoproteins (HDL). Therefore, many researches focused on modifying the egg’s fatty acids profile in favor of unsaturated fatty acids, namely ω-3 PUFA [40,41,42,43]. Indeed, it has been shown that ω-3 PUFAs such as α-linolenic acid (ALA, C18:3), eicosapentaenoic acid (EPA, C20:5) and docosahexaenoic acid (DHA, C22:6), have many health benefits namely with regard to providing a primary protection against cardiovascular diseases [44].

Eggs can be fortified with ω-3 PUFAs by supplementing the feed of the laying hens with a source of ω-3 PUFA. Dietary ω-3 PUFAs are then transferred to the eggs [45,46,47,48]. Linseed contains about 34% oil and has a high content of ALA (>50%) [49], which makes it a good source for the ω-3 PUFA enrichment of egg yolk [50,51,52,53].

Unsaturated fatty acids are highly susceptible to oxidation. Consequently, the incorporation of PUFA in the layer feed through linseed incorporation may increase the susceptibility of eggs’ lipids to oxidation [54]. Antioxidant substances may be added to the layer feed to protect fatty acids from oxidation [11]. Tomatoes and red pepper are both important sources of natural antioxidants and carotenoids, and it has been reported that the prevention of cholesterol oxidation and, presumably, of PUFA auto-oxidation may be achieved by adding tomatoes [55] or sweet red pepper [56] into hens’ feed tomatoes.

The consumption of fatty acids can have a direct effect on the stimulation or preclusion of atherosclerosis and coronary thrombosis due to their effect on blood cholesterol and low-density lipoprotein (LDL) cholesterol concentrations [57]. Accordingly, the index of atherogenicity (IA) and index of thrombogenicity (IT) have been introduced [57]. The IA and the IT might better characterize the atherogenic and thrombogenic potential of the egg yolk lipid than simple approaches such as PUFA/SFA ratio determination (where SFA means saturated fatty acids). Indeed, IA and IT take into account the different effects that single fatty acids might have on human health (i.e., the consumer) and particularly on the probability of increasing the incidence of pathogenic phenomena, such as atheroma and/or thrombi formation. Indeed, eggs with a lower SFA/UFA ratio showed low values of IA, IT, and hypercholesterolemic (HH) indexes, and they have been recommended for a healthy diet [57,58].

In a previous work [59], it has been shown that the inclusion of linseed at a level of 4.5% in the laying hen diet was associated with a significant enrichment of eggs with carotenoids and with an enhancement of their antioxidant status. However, this beneficial response was not improved by the simultaneous enrichment of hens’ feed with linseed (4.5% of the feed), as a source of ω-3 polyunsaturated fatty acids, and a tomato and red pepper mix (2% of the feed), as natural antioxidants. In our previous paper, the egg yolk fatty acids profile and health lipid indexes were not reported. Thus, the objective of the present study is to evaluate the effect of the dietary incorporation of linseed alone or along with a dried tomato paste-pepper powder mix on laying hens’ egg yolk fatty acids profile and health lipid indexes.

## 2. Materials and Methods

### 2.1. Ethical Considerations

All procedures related to animals’ care, handling, and sampling were conducted under the approval of the Official Animal Care and Use Committee of the Higher School of Agriculture of Mateur (protocol N°05/15) before the initiation of research, and they followed the Tunisian official guidelines.

### 2.2. Experimental Design

Sixty (60) 27-week-old Novogen White laying hens (initial live body weight = 1449.95 g ± 71.99 g) were divided into 3 homogeneous groups of 20 hens each. A standard mash diet (control diet) for laying hens based on corn and soybean-meal was prepared. Thereafter, 2 supplemented diets, designated as follows: (1) linseed (L), and (2) linseed-tomato-pepper (LTP), were individually prepared by mixing the control diet thoroughly with the designated supplement at the required incorporation level as described in our previous research study [59]. The ingredients and chemical composition of the three diets are given in Table 1.

With respect to the fatty acids composition, the control feed (C) contained a relatively high level of palmitic acid (C16:0) as compared to the linseed-supplemented feeds L and LTP. Linseed-supplemented feeds (L and LTP) had a high percentage of α-Linolenic (ALA, C18:3) up to 17.27% and 19.95%, respectively. These percentages were almost 18 times higher than the one measured for the control feed (1.13%). The same trend was observed in the percentage of ω-3 PUFA.

The diets were restricted to 100 g/hen/day to ensure that the hens consumed almost all of the distributed feed and to reduce the feed-selection behavior typically observed in laying hens. The feed was offered once daily at 7:30 am. The hens were housed individually in standard pens with an individual feed-trough and common water-trough. The ambient temperature was about 20 ± 4 °C. The hens were maintained under a schedule of 16 h light:8 h darkness. The drinking water was supplied *ad libitum* throughout the duration of the experiment (47 days).

### 2.3. Egg Yolks Sampling and Processing

Sixty (60) eggs laid on the 31st day of the experimental period were weighted. The eggs’ weight varied between 51 and 61 g. The eggs were broken, and yolk and white were separated. The egg yolks were then pooled per two hens belonging to the same dietary treatment group, so that 10 yolk samples per group were obtained instead of 20. Thereafter, the pooled yolk samples (*n* = 10) were recovered in previously tared opaque containers, stored at −18 °C for 48 h and subsequently lyophilized for 24 h. The resulting freeze-dried yolk samples were weighted, pulverized into a fine powder in a mill and stored in closed, dry, opaque containers, away from the light until the yolk fatty acid composition determination.

### 2.4. Chemical Analysis

The total fat content and fatty acids profiles were determined on the lyophilized yolk samples in the laboratories of the Department of Pharmacy, at the University of Napoli Federico II, Napoli (Italy), following the methodology reported by [60].

#### 2.4.1. Egg Yolk’s Fat Extraction

0.5 g of lyophilized egg yolk were weighted and mixed with 7.5 mL of methanol. The mixture was then homogenized with Ultrax-Turax (Ika^®^ digital Ultra-Turax) at 6000 rpm for 5 min. Then, 3.75 mL of chloroform were added. After agitation for 10 min, 2.25 mL of a salt solution (2.9 g NaCl + 0.2 g of CaCl_2_/L) were then added. After centrifugation (Thermo Scientific ‘SL16R’) for 15 min (5000 rpm, 20 °C), the supernatant was transferred into a 50 mL centrifuge tube. The precipitate was re-extracted with 3.75 mL of chloroform. The mixture was vigorously agitated for 10 min and centrifuged for 10 min (5000 rpm, 20 °C). The supernatant was added to the first one and 3.75 mL of the salt solution were added. The lower phase was isolated, made up to 12.5 mL with methanol, before being transferred into a previously tared flask and reduced to dryness by evaporating the solvent in a rotating evaporator (200 rpm, 40 °C). After the complete removal of the solvent, the flask was weighted and the amount of extracted fat was calculated by subtracting the weight of the tare.

#### 2.4.2. Preparation of Fatty Acid Methyl Ester

One hundred milligrams of extracted fat were weighted and solubilized in 2 mL of n-hexane. Aliquots of 1 mL of the obtained solution were trans-esterified by mixing with 300 µL of potassium hydroxide 2 N in methanol using a Vortex stirring device for 1 min. Thus, two phases were obtained. Thereafter, 800 µL of the upper phase were transferred into a glass vial.

#### 2.4.3. Gas Chromatography Analysis of Fatty Acids

The analysis of fatty acid methyl esters was carried out using a Shimadzu 17A gas chromatograph equipped with a fused silica capillary column (Phenomenex ZB-Wax, 0.50 µm film thickness, 60 m × 0.32 mm id) and a Flame Ionization Detector Helium was used as the carrier gas at 2 mL min-1. The temperature of the program was 200 °C for 5 min, 200 °C up to 230 °C for 15 min (2 °C/min), and constant at 230 °C for 30 min. The column injector and Flame Ionization Detector (FID) temperatures were 200, 240 and 240 °C, respectively. The fatty acids were identified from their peak retention times compared to those of the known standards (Merck, Darmstadt, Germany).

### 2.5. Lipid Health Quality Indexes Calculation

The health quality of the egg yolks’ lipids was assessed by calculating: (1) the index of atherogenicity (IA), (2) the index of thrombogenicity (IT) and (3) the ratio between the hypocholesteronic and hypercholesteronic (HH) fatty acids.

The index of atherogenicity (IA) indicates the relationship between the sum of the main saturated fatty acids and that of the main classes of unsaturated ones. Saturated fatty acids are considered pro-atherogenic (favoring the adhesion of lipids to cells of the immune and circulatory system). However, unsaturated fatty acids are qualified as anti-atherogenic (inhibiting the aggregation of plaque and diminishing the levels of esterified fatty acid, cholesterol, and phospholipids, thereby preventing the appearance of micro- and macro-coronary diseases).

The index of thrombogenicity (IT), defined as the ratio of the pro-thrombogenetic (saturated) to the anti-thrombogenetic (unsaturated) fatty acids, shows the tendency to form clots in the blood vessels.

The following equations were used to calculate these indexes:

- Atherogenic Index [57]

IA = (4 × C14:0 + C16:0 + C18:0)/(ΣMUFA + ΣPUFA-ω-6 + ΣPUFA-ω-3)

- Thrombogenic Index [57]

IT = (C14:0 + C16:0 + C18:0)/(0.5 × ΣMUFA + 0.5 × ΣPUFA-ω-6 + 3 × ΣPUFA-ω-3 + ΣPUFA-ω-3/ΣPUFA-ω-6)

- Ratio between hypocholesterolemic and hypercholesterolemic fatty acids [58]

HH = (C18:1ω-9 + C18:2ω-6 + C20:4ω-6 + C18:3ω-3 + C20:5ω-3 + C22:5ω-3 + C22:6ω-6)/(C14:0 + C16:0)

where Σ = Summatory, MUFA = monounsaturated fatty acids, and PUFA = polyunsaturated fatty acids.

### 2.6. Statistical Analysis

The egg yolk fatty acids data were tested for treatment effect using the General Linear Model (GLM) procedure of the Statistical Analysis System SAS [61], according to the following model:
*Yij* = µ + *Ti* + *eij*
where:
*Yij* = represents the *j*^th^ observation on the *i*^th^ treatmentµ = overall mean*Ti* = the main effect of the *i*^th^ treatment *eij* = random error present in the *j*^th^ observation on the *i*^th^ treatment.


## 3. Results and Discussion

### 3.1. Egg Yolk Fatty Acids Profile

The fatty acids composition of the egg yolk of hens fed on tested feeds is given in Table 2.

The data showed that there were no statistical differences in the egg yolk fat content between the 3 dietary groups (*p* > 0.05). In fact, the total fat percent percentage ranged from 39.85% DM (control) to 46.15% DM (L). This finding was in agreement with that of Imran et al. [62] who reported that the dietary addition of 10, 20 and 30% of extruded linseed did not affect the egg yolk total lipids content.

Regardless of the dietary treatment, palmitic acid (C16:0), oleic acid (C18:1ω-9) and linoleic acid (C18:2ω-6) were the most abundant fatty acids found in the egg yolks. This has been also confirmed by other authors [63,64,65,66].

In the present study, the linseed inclusion in hens’ feed at a level of 4.5% reduced (*p* < 0.05) the egg yolk palmitic acid (C16:0) concentration from 25.41% (C) to 23.43% (L). This result is consistent with the data reported by Sari et al. [53], Criste et al. [67] and Yassein et al. [68], who demonstrated that the dietary incorporation of 5% of linseed resulted in a decrease in the egg yolk content of palmitic acid. Palmitic acid decreased from 23.45% to 21.47% [53], and from 28.74% to 26.39% [68]. The egg yolk stearic acid (C18:0) concentration also decreased (*p* < 0.05) from 14.75% (C) to 12.52% (L). In this regard, a decrease in this saturated fatty acid has been reported by Yalcin et al. [69], in particular from 9.01% to 9%, as well as by Yassein et al. [68] who observed a decrease from 11.79% to 10.60%. On the other hand, the dietary addition of linseed resulted in an increase in the oleic acid (C18:1) level from 24.39% to 26.9%. This observation is in accordance with previous reports [53,67,68,69]. Oleic acid increased from 38.66% to 37.12% [53] and from 37.08 % to 42.79 % [68].

Logically, the use of linseed in layer feed should increase the egg yolk content of unsaturated fatty acids, namely that of the linolenic acid, as well as those of the eicosapentaenoic acid (EPA) and the doxosahexaenoic acid (DHA), which are the metabolic derivatives of the linolenic acid. Indeed, linseed has a high content of α-Linolenic (C18:3, ALA) fatty acid (>50%) [49]. Thus, an increase in the dietary ALA content should increase that of the egg yolks. Moreover, in the liver, birds are capable of converting ALA into two main long chain omega-3 fatty acids, eicosapentaenoic acid (EPA) and docosahexaenoic acid (DHA). These are transferred to the egg yolk in small amounts.

In the current study, the inclusion of 4.5% linseed in the feed did not affect the α-Linolenic (C18:3, ALA) concentration in yolks, although the content of this fatty acid has increased considerably in the feed from 1.13% for the control to 17.27% for the linseed-supplemented feed. Only a numerical increase of 1.43% was recorded in the yolk ALA content from 0.34% (C) to 1.77% (L). Yassein et al. [68] observed an increase of 0.54% in the ALA amount in egg yolk of layers fed on a diet which contained 5% of linseed. However, other studies [69,70] using linseed dietary incorporation rates very close (5% and 4.32%) to that used in this study (4.5%), reported an increase of up to 3.18% and 2.87%, respectively, larger than that recorded in this paper.

The reasons for the low dietary ALA transfer to the yolk lipids of hens fed 4.5% linseed in the current research are not evident. It could be due to the age of the hens. It has been reported that the absorption rate of ALA increases with the age of hens [63,71]. This hypothesis is quite plausible since, in previous works [69,70], egg yolks in which the concentration of ALA was determined were respectively laid by hens 4 weeks and 7 weeks older than those used in the current experiment. Moreover, Yannakopoulos et al. [35] studied the effect of the dietary addition of flaxseed on the egg yolk fatty acids composition from the 24th week of age to the 60th week of age. They found that the egg yolk concentration of linolenic acid, eicosapentaenoic acid (EPA), and doxosahexaenoic acid (DHA) increased from the 24th week to the 32nd week of age and remained constant until the 55th week. Even if a recent study of Ehr et al. 2017 [72] that focused on evaluating the transfer of ALA, EPA, and DHA into egg yolk from extracted flaxseed oil or milled flaxseed in Hy-Line W-36 laying hens over an 8-week feeding period (25 to 33 weeks old) found that dietary flaxseed oil improved the feed efficiency and increased the ALA deposition into yolk compared to a milled source, other factors should also be considered, and further studies should be carried out.

Furthermore, in a study carried out by Dalle Zotte et al. [73], wherein egg yolks from hens fed a ground linseed-supplemented diet and hens fed an extruded linseed-supplemented diet were compared, it was found that linseed processing influenced the deposition levels of ω-3 PUFA, ALA, and EPA in egg yolks. This finding seems to suggest that the extruded form of linseed led to a higher and faster absorption of dietary PUFA when compared to the ground form, especially with regard to *n*-3 PUFA, ALA, and EPA. The authors tried to explain their finding by the fact that the extruded form was deprived of anti-nutritional factors and became more readily digestible, as reported by Thacker et al. [74].

In the research reported here, the linseed diet containing 17.27% of ALA was associated with a yolk lipid ALA content of 1.77%. The same feed containing 48.56% of LA induced an average yolk lipid LA content of 22.97%. Thus, the dietary LA is more efficiently incorporated into egg yolk than ALA.

EPA and DHA are the most potent *n*-3 polyunsaturated fatty acids in relation to human health [75]. The WHO recommends a 300 to 500 mg daily intake of EPA and DHA. In the present study, the EPA and DHA concentrations of yolks increased significantly (*p* < 0.05) when 4.5 % of linseed was included in the feed. Previous studies also reported an increase in the concentration of these two ω-3 PUFA in the yolk lipids from hens fed increasing levels of linseed from 5 to 15% [53] or fed 4.32% of linseed [69].

The EPA concentration of yolk lipids observed in our experiment corresponds closely to the findings of Sari et al. [53] (EPA = 0.08%), and to those of Yalcin et al. [69] (EPA = 0.06%), when they incorporated linseed into the feed at a level of 5%.

The DHA level in egg yolk laid by hens that were fed a linseed-supplemented feed was similar to that reported by Sari et al. [53] when hens were fed 5% of linseed (2.21%) and higher than that found by Yalcin et al. [69] (1.70%).

Hens fed linseed feed containing 17.27% of ALA produced egg yolks containing 0.05% of EPA and 2.73% of DHA. The EPA level remains much lower than that of DHA. Consequently, it can be concluded that the conversion of dietary ALA into EPA, as based on the fatty acid composition of eggs, was not efficient. It is likely that this conversion was inhibited by the high intake of LA. In fact, in the present study, linseed-supplemented feed had a higher content of LA (48.56%) than that of ALA (17.27%). ALA is the major ω-3 fatty acid metabolized to EPA and DHA, while LA is the major ω-6 fatty acid metabolized to AA [76]. ALA and LA compete with each other for the same enzymes involved in the desaturation and elongation reactions [77,78]. Additionally, it is well known that the LA:ALA ratio has a role in the conversion of ALA into EPA and DHA. In this regard, it has been shown that a reduced ratio will encourage a longer chain PUFA synthesis [41].

When we calculated the concentrations of ALA, EPA, and DHA per egg, we found that one egg from a hen fed a diet containing 4.5% linseed would provide about 88.5 mg of ALA, 2.35 mg of EPA and 135.7 mg of DHA. The EPA and DHA amounts would supply about 28 to 46% of the daily-recommended intake (300 to 500 mg of EPA and DHA). Since EPA and DHA are present only in fish products, eggs enriched with EPA and DHA could be a good source of these ω-3 polyunsaturated fatty acids for people not consuming fish products.

In the present study, tomato and sweet pepper were added to the linseed supplemented feed at an incorporation level of 1% each in order to prevent egg yolk lipids oxidation. Indeed, according to our previous report [59], tomato and red pepper are very rich in carotenoids, displaying an important antioxidant activity. In the present work, the addition of a dried tomato paste and sweet red pepper mix to the linseed feed (LTP) did not influence the egg yolk fatty acids profile compared with the linseed feed (L). This result seems quite logical, since the main goal of the present study was to produce eggs with functional properties by the incorporation of linseed, as a source of ω-3 polyunsaturated fatty acids, as well as the incorporation of tomato and pepper, as sources of natural antioxidant, into laying hens’ diets. However, it was interesting for us to also investigate the hypothesis that the fatty acid composition of egg yolk could be affected by the dietary inclusion of tomato and sweet pepper, since in a previous study it has been reported that different levels of dietary antioxidant (vitamin E) slightly affected the fatty acid composition of the yolk [79].

### 3.2. Lipid Health Quality Indexes

Egg yolk fatty acids have been grouped in classes according to their number of double bonds as saturated (SFA), monounsaturated (MUFA), and polyunsaturated (PUFA) fatty acids. These classes have been used to define some lipid health quality indicators like the ratio between hypocholesterolemic and hypercholesterolemic fatty acids (HH), as well as the atherogenic (IA) and thrombogenic (IT) indexes. The data for the sum of fatty acids per class, ω6:ω3 ratio and lipid health indexes of eggs’ yolks are shown in Table 3.

The sum of SFA, PUFA, ω3-PUFA and the ω6-PUFA:ω3-PUFA ratio were affected (*p* < 0.05) by the dietary treatment. Egg yolks that were derived from hens that were fed linseed-supplemented feeds (L and LTP) had a lower SFA content and ω6-PUFA:ω3-PUFA ratio than those laid by the control hens, but had higher levels of PUFA and ω3-PUFA. These results are in line with those found by Sari et al. [53]. Indeed, these authors reported that the dietary incorporation of 5% of linseed reduced the level of SFA from 32.28% to 30.45%. Yassein et al. [68] also found that linseed supplementation at 5% reduced the egg yolk concentration of SFA from 40.91% to 37.12% and increased the total content of PUFA from 55.52% to 62.90%.

Feeding linseed-supplemented diets (L and LTP) did not affect the IA and the HH indexes but significantly reduced the IT. It has been reported that animal products with a low index of thrombogenicity decrease the threat of atrial fibrillation [80].

Although variations among the three dietary groups regarding the HH index were not significant, those calculated for egg yolks derived from hens that were fed linseed-supplemented diets (L and LTP) were numerically higher than that for control egg yolks. According to Santos-Silva et al. [58], the higher the ratio between hypocholesterolemic and hypercholesterolemic fatty acids, the more the oil or fat is adequate to human nutrition.

It is difficult to compare the values of the IA, IT, and HH indexes obtained in this research for egg yolks laid by hens that were fed linseed-supplemented diets to those of other studies because, in the literature, there is a lack of work related to the assessment of indexes of hens’ egg yolk following linseed dietary incorporation.

## 4. Conclusions

The results of this study clearly demonstrate that the dietary inclusion of 4.5% of linseed significantly reduced the content of SFA and the ω6-PUFA:ω3-PUFA ratio and increased the PUFA contents, namely the eicosapentaenoic EPA) and doxosahexaenoic acid (DHA) content, in laying hens’ egg yolks; it also notably decreased their thrombogenic index. The simultaneous enrichment of hens’ feed with linseed and natural antioxidants (from tomato and red pepper) did not enhance the dietary quality of the egg yolk fatty acids but it might be useful in preventing egg yolks’ polyunsaturated fatty acids oxidation.

## Figures and Tables

**Table 1 nutrients-11-00813-t001:** Ingredients, and chemical and fatty acids composition of the experimental diets *.

	Diets		
	Control (C)	Linseeds (L)	Linseeds-Tomato-Pepper (LTP)
**Ingredients (%)**			
Linseed	0	4.5	4.5
Fenugreek seed	0	0	0
Dried Tomato	0	0	1
Sweet red pepper	0	0	1
Yellow corn	66.5	63.5	61.5
Soybean meal	25.5	24	24
Calcium carbonate, Mineral and Vitamin mixture ^α^	8	8	8
**Chemical Composition**			
Crude protein, (%, DM)	18.1	18	18
Ether extract (%, DM)	3.56	5.6	5.27
Metabolizable energy (kcal/kg DM)	2750	2850	2830
**Fatty acids composition (% of total fat)**			
Palmitic (C16:0)	14.13	12.80	11.61
Stearic (C18:0)	3.81	5.72	5.39
Oleic (C18:1)	22.55	11.33	16.29
Linoleic (C18:2)	51.69	48.56	47.22
α-Linolenic (C18:3)	1.13	17.27	19.95
∑ SFA	18.81	19.97	18.78
∑ PUFA-ω3	1.13	17.17	19.95

***** C = Control diet; L = diet supplemented with ground linseed at 4.5%, LTP = diet supplemented with ground linseed (4.5%), dried tomato paste (1%) and sweet pepper powder (1%) mix; ΣSFA: sum of saturated fatty acids; ΣMUFA: sum of monounsaturated fatty acids; ΣPUFA: sum of polyunsaturated fatty acids; ΣPUFA-ω6 = C18:2ω- 6 + C20:2ω-6 + C20:4ω-6; ΣPUFA-ω3 = C18:3ω-3 + C20:3ω-3 + C20:5ω-3 + C22:6ω-3; DM = Dry Matter. ^α^ Control (C) provided following nutrients per 100 g: Ca, 4.3 g; P, 0.6 g; Na, 0.14 g; Cl, 0.23 g; Fe, 4 mg; Zn, 40 mg; Mn, 7 mg; Cu, 0.3 mg; I, 0.08 mg; Se, 0.01 mg; Co, 0.02 g; methionine, 0.39 g; methionine + cysteine, 0.69 g; lysine, 0.89 g; Retinol, 800 IU; Cholecalciferol, 220 IU; α-tocopherol, 1.1 IU; Thiamin, 0.33 IU; Nicotinic acid, 909 IU.

**Table 2 nutrients-11-00813-t002:** Fatty acids composition (% of total fat) of egg yolk ^$^.

Item			Diets		
	C	L	LTP	SEM ^γ^	P
Total fat, % DM	39.85	46.15	43.05	2.11	NS
Pentadeconoic (C15:0)	0.08	0.06	0.07	0.007	NS
Palmitic (C16:0)	25.41 ^a^	23.43 ^b^	23.6 ^b^	1.06	*
Margaric (C17:0)	0.23	0.21	0.22	0.02	NS
Stearic (C18:0)	14.75 ^a^	12.53 ^b^	11.85 ^b^	0.68	*
Nonadecanoic (C19:0)	0.04	0.03	0.03	0.004	NS
Arachidic (C20:0)	0.05 ^a^	0.04 ^b^	0.041 ^b^	0.003	*
Palmitoleic (C16:1)	4.16 ^a^	3.28 ^b^	3.38 ^b^	0.22	NS
Hexadicadienoic (C16:2)	0.03	0.024	0.02	0.002	NS
Hexadicatrienoic (C16:3)	0.01	0.01	0.014	0.003	NS
Oleic (C18:1)	24.4 ^b^	26.9 ^a^	26.68 ^a^	1.82	*
Linoleic (C18:2)	20.70	20.44	20.03	0.02	NS
α-Linolenic (C18:3)	0.34 ^b^	1.77 ^a^	1.89 ^a^	0.8	NS
Nonadecenoic (C19:1)	0.025 ^b^	0.043 ^a,b^	0.044 ^a,b^	0.005	*
Gadoleic (C20:1)	0.4	0.33	0.29	0.04	NS
Eicosadienoic (C20:2)	0.21	0.19	0.2	0.03	NS
Eicosatrienoic (C20:3)	0.57	0.45	0.41	0.04	NS
Arachidonic (C20:4)	2.6	2.28	2.29	0.49	NS
Eicosapentanoic (C20:5)	0.01 ^b^	0.05 ^a^	0.033 ^a,b^	0.06	*
Docosahexaenoic (C22:6)	1.94 ^b^	2.73 ^a^	2.66 ^a^	0.18	*

^$^ C = Control diet; L = diet supplemented with ground linseed at 4.5%, LTP = diet supplemented with ground linseed (4.5%), dried tomato (1%) and sweet pepper powder (1%) mix; ^γ^ SEM = standard error of the mean; *p*-values < 0.05 were considered significant: * = *p* < 0.05, NS = *p* ≥ 0.05, NS = Not Significant. ^a,b^: Means within the same row with different superscript letters are significantly different (*p* < 0.05).

**Table 3 nutrients-11-00813-t003:** Sum and ratios of fatty acids (%) and health lipid indexes of lyophilized eggs’ yolks ^$^.

			Diets		
	C	L	LTP	SEM ^γ^	P
∑ SFA	40.51 ^a^	36.28 ^b^	35.81 ^b^	1.10	*
∑ MUFA	28.38	30.59	30.39	1.52	NS
∑ PUFA	26.41 ^b^	27.94 ^a^	27.56 ^a^	1.4	*
∑ PUFA-ω6	23.51	22.91	22.52	0.7	NS
∑ PUFA-ω3	2.86 ^b^	4.99 ^a^	5.0 ^a^	0.25	*
∑ω-6/∑ω-3	8.22 ^a^	4.59 ^b^	4.5 ^b^	1.82	*
IA	0.74	0.69	0.61	0.07	NS
HH	1.92	2.41	2.36	0.91	NS
IT	1.16 ^a^	0.86 ^b^	0.85 ^b^	0.15	*

**^$^** C = Control diet; L = diet supplemented with ground linseed at 4.5%, LTP = diet supplemented with ground linseed (4.5%), dried tomato (1%) and sweet pepper powder (1%) mix; ΣSFA: sum of saturated fatty acids; ΣMUFA: Sum of monounsaturated fatty acids; ΣPUFA: sum of polyunsaturated fatty acids; ΣPUFA-ω6 = C18:2ω-6 + C20:2ω-6 + C20:4ω-6; ΣPUFA-ω3 = C18:3ω-3 + C20:3ω-3 + C20:5ω-3 + C22:6ω-3; ^γ^ SEM = Standard error of the mean; *p*-values < 0.05 were considered significant: * = *p* < 0.05, NS = *p* ≥ 0.05, NS = Not Significant. ^a,b^: Means within the same row with different superscript letters are significantly different (*p* < 0.05).

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
