# Peer review of "Effect of Dietary Incorporation of Linseed Alone or Together with Tomato-Red Pepper Mix on Laying Hens’ Egg Yolk Fatty Acids Profile and Health Lipid Indexes"

_nutrients, 2019, doi:10.3390/nu11040813_

Round 1
Reviewer 1 Report
The manuscript was properly conducted, and findings reported are important for poultry production and products quality. Moreover, the Authors investigated an interesting topic, and the objective of the paper is of worldwide interest and fits well within the overall scope of the journal. Results were properly reported, and the findings have been accurately discussed and compared with other recently published papers.
Author Response
Review Report Form
Open Review
(x) I would not like to sign my review report
( ) I would like to sign my review report
English language and style
( ) Extensive editing of English language and style required
( ) Moderate English changes required
( ) English language and style are fine/minor spell check required
(x) I don't feel qualified to judge about the English language and style
Yes | Can be improved | Must be improved | Not applicable | |
Does the introduction provide sufficient background and include all relevant references? | (x) | ( ) | ( ) | ( ) |
Is the research design appropriate? | (x) | ( ) | ( ) | ( ) |
Are the methods adequately described? | (x) | ( ) | ( ) | ( ) |
Are the results clearly presented? | (x) | ( ) | ( ) | ( ) |
Are the conclusions supported by the results? | (x) | ( ) | ( ) | ( ) |
Comments and Suggestions for Authors
The manuscript was properly conducted, and findings reported are important for poultry production and products quality. Moreover, the Authors investigated an interesting topic, and the objective of the paper is of worldwide interest and fits well within the overall scope of the journal. Results were properly reported, and the findings have been accurately discussed and compared with other recently published papers.
The authors want to express their gratitude to the reviewer for his/her valuable comments and suggestions. The authors’ replies to the individual points raised are reported in Italic below.
Submission Date
21 February 2019
Date of this review
28 Feb 2019 01:29:07

Reviewer 2 Report
There are multiple English-language, spelling, and spacing errors throughout the document and as such, it is not acceptable for publication in this current state. There are places where too many spaces are used, and others where there are no spaces between 2 words. Some words remain in french spelling style where the rest is English. There are too many to list individually and it retracts from the message of the paper. The authors are encouraged to solicit a native English speaker for editing.
Table 1- The listed ingredients do not seem to provide a balanced diet as written, and this table needs more description before publication. Are all amino acid, calcium, and phosphorus needs for a hen in lay met? It is not clear from table as presented.
How did you arrive at 1% inclusion rate for the anti-oxidative compounds? And why? This is not clear.
There is an issue with feeding diets only at a rate of 100g/day, and energy values being vastly different. Why were the diets not balanced based on ME?
These laying hens are similar in age, and an increased deposition was seen over time with ground flaxseed. Why do you think your results differ? https://www.ncbi.nlm.nih.gov/pubmed/28108729
I do not see egg production reported, or feed efficiency/egg mass. If this is expected to be used commercially, these are important values to report. What happened to the BW of the hens over time? What about egg color index?
Line 147 What is the point of calculating the lipid health indices? This is not well-explained. If this is something you advocate for so others include in their work, it needs to be better explained and justified.
Line 167 General Linear Model (GLM) does not describe the actual methods used for statistical analysis. What were the factors used and error accounted for? Main effects, interactions? This section is not descriptive enough to understand what you actually did for the analysis.
What measures of oxidation or oxidative capacity did you measure in the egg yolk? It is not clear in the background or results why the third diet was included.
There are several more descriptive analyses needed for the eggs, hen health, production, and dietary inputs needed before this paper is publishable. This is needed for publication.
The results and discussion need to be more quantitative. There are some sections that report numbers well, but when comparing results, it is not ok to say something simply increased or decreased. The reader has to then search for numeric results in the tables. What is the percent change? What represents biological significance?
Line 315- why would these compounds affect egg yolk fatty acid?
Line 316- did you measure oxidation? How can you come to this conclusion if you did not?
Author Response
The authors want to express their gratitude to the reviewer for his/her valuable comments and suggestions. The authors’ replies to the individual points raised are reported in Italic below.
Review Report Form
Open Review
(x) I would not like to sign my review report
( ) I would like to sign my review report
English language and style
(x) Extensive editing of English language and style required
( ) Moderate English changes required
( ) English language and style are fine/minor spell check required
( ) I don't feel qualified to judge about the English language and style
The linguistic revision of whole manuscript was carried out.
Yes | Can be improved | Must be improved | Not applicable | |
Does the introduction provide sufficient background and include all relevant references? | ( ) | ( ) | (x) | ( ) |
Is the research design appropriate? | ( ) | ( ) | (x) | ( ) |
Are the methods adequately described? | ( ) | ( ) | (x) | ( ) |
Are the results clearly presented? | ( ) | ( ) | (x) | ( ) |
Are the conclusions supported by the results? | ( ) | ( ) | (x) | ( ) |
Comments and Suggestions for Authors
There are multiple English-language, spelling, and spacing errors throughout the document and as such, it is not acceptable for publication in this current state. There are places where too many spaces are used, and others where there are no spaces between 2 words. Some words remain in french spelling style where the rest is English. There are too many to list individually and it retracts from the message of the paper. The authors are encouraged to solicit a native English speaker for editing.
The linguistic revision of whole manuscript was carried out. This remark was taken into consideration and the necessary corrections were made. References were checked . The title of two references were in French.
1) Table 1- The listed ingredients do not seem to provide a balanced diet as written, and this table needs more description before publication. Are all amino acid, calcium, and phosphorus needs for a hen in lay met? It is not clear from table as presented.
Yes, all amino acids, calcium and phosphorus needs were met. A note was inserted with major details.
2) How did you arrive at 1% inclusion rate for the anti-oxidative compounds? And why? This is not clear.
How: 1% equal to 1 kg of anti-oxidative compounds in 100 Kg of basal diet (61.5 Kg of yellow corn, 24 Kg of soy bean meal, 1 kg of red pepper powder and 1 Kg of dried tomato).
Why: In Akdemir et al. (2012) tomato powder added at an inclusion levels of 0.5 and 1% enhanced linearly laying performance and increased concentrations of egg yolk lycopene, β-carotene, and lutein and decreased MDA concentrations. With regard to red pepper dietary supplementation effect, red pepper powder used at level of 0.8% did not affect egg-laying performance, feed consumption and FCR but increased the yolk color score (Li et al., 2012).
3) There is an issue with feeding diets only at a rate of 100g/day.
The explanation of the authors is given in lines 135-136
4) Energy values being vastly different. Why were the diets not balanced based on ME?
Dietary addition of flaxseeds increased ether extract percent and consequently the calculated metabolizable energy. And, according to the management guide of the Novogen white strain, this laying hen (from laying rate of 5% to the age of 50 weeks) can receive a feed with 2750-2900 Kcal/Kg of ME without any negative effect on its laying performances.
5) These laying hens are similar in age, and an increased deposition was seen over time with ground flaxseed. Why do you think your results differ? https://www.ncbi.nlm.nih.gov/pubmed/28108729
Supplementation of flaxseed in layer feed should increase the egg yolk content of unsaturated fatty acids, namely that of linolenic acid, as well as that of eicosapentaenoic acid (EPA) and doxosahexaenoic acid (DHA) (metabolic derivatives of the linolenic acid). Concerning laying hens’ age, Yannakopoulos et al. (2005) studied the effect of dietary addition of flaxseed on egg yolk fatty acids composition from the 24th week of age to the 60th week of age. They found that egg yolk concentration of linolenic acid, eicosapentaenoic acid (EPA) and doxosahexaenoic acid (DHA) increased from the 24th week to the 32nd week of age and remained constant until the 55th week.
Moreover, as you suggested, the proper and recent example, was described in the text. Also in the text was underlined how also other factors should be considered and further studies carried out.
6) I do not see egg production reported, or feed efficiency/egg mass. If this is expected to be used commercially, these are important values to report. What happened to the BW of the hens over time? What about egg color index?
The research described in this article is a continuation of that presented in previous publication (Omri, B.; Chalghoumi, R.; Abdouli, H. Study of the Effects of Dietary Supplementation of Linseed, Fenugreek Seeds and Tomato-Pepper Mix on Laying Hen’s Performances, Egg Yolk Lipids and Antioxidants Profiles and Lipid Oxidation Status. J. Anim. Sci. Livestock. Product. 2017, 1:2.) in which laying performances traits were presented and discussed.
7) Line 147 What is the point of calculating the lipid health indices? This is not well-explained. If this is something you advocate for so others include in their work, it needs to be better explained and justified.
This remark was taken into consideration. Please see lines 86-95
Line 167 General Linear Model (GLM) does not describe the actual methods used for statistical analysis. What were the factors used and error accounted for? Main effects, interactions? This section is not descriptive enough to understand what you actually did for the analysis.
In the present study we evaluated the effect of dietary incorporation of linseed alone or together with tomato-red pepper mix (treatment) on laying hens’ egg yolk fatty acids profile and health lipid indices (main effects). There is only one factor used in our study, which was the treatment so we cannot speak about interaction between two or many factors.
8) What measures of oxidation or oxidative capacity did you measure in the egg yolk?
These analytic measures are well described in our previous paper (Omri, B.; Chalghoumi, R.; Abdouli, H. Study of the Effects of Dietary Supplementation of Linseed, Fenugreek Seeds and Tomato-Pepper Mix on Laying Hen’s Performances, Egg Yolk Lipids and Antioxidants Profiles and Lipid Oxidation Status. J. Anim. Sci. Livestock. Product. 2017, 1:2.)
9) It is not clear in the background or results why the third diet was included.
Please see lines 96-101 and lines 377-381
There are several more descriptive analyses needed for the eggs, hen health, production, and dietary inputs needed before this paper is publishable. This is needed for publication.
As stated above, the research described in this manuscript is a continuation of that presented in previous published work (Omri, B.; Chalghoumi, R.; Abdouli, H. Study of the Effects of Dietary Supplementation of Linseed, Fenugreek Seeds and Tomato-Pepper Mix on Laying Hen’s Performances, Egg Yolk Lipids and Antioxidants Profiles and Lipid Oxidation Status. J. Anim. Sci. Livestock. Product. 2017, 1:2.) in which eggs, hen health, production, and dietary inputs traits were described and discussed.
10) The results and discussion need to be more quantitative. There are some sections that report numbers well, but when comparing results, it is not ok to say something simply increased or decreased. The reader has to then search for numeric results in the tables. What is the percent change? What represents biological significance?
This remark was taken into consideration throughout the “results and discussion” section
11) Line 315- why would these compounds affect egg yolk fatty acid?
The explanation of the authors is given in lines 377-381
Line 316- did you measure oxidation? How can you come to this conclusion if you did not?
As mentioned above, this study is the second part of an already published work (Omri et al. 2017). In that study, authors have investigated the effect of dietary Supplementation of tomato-pepper antioxidants profiles and egg yolk lipid oxidation status.
Submission Date
21 February 2019
Date of this review
17 Mar 2019 17:13:03

Round 2
Reviewer 2 Report
Paper still needs reviewing by a native english speaker.
For example:
"Even if, a recent study of Ehr et al. 2017, by evaluating the transfer of ALA, EPA"
"In fact, Total fat percent varied from"
These are not the only occurrences, just examples.
Author Response
The authors want to express their gratitude to the reviewer for his/her valuable comments.
Open Review
(x) I would not like to sign my review report
( ) I would like to sign my review report
English language and style
( ) Extensive editing of English language and style required
(x) Moderate English changes required
( ) English language and style are fine/minor spell check required
( ) I don't feel qualified to judge about the English language and style
Yes | Can be improved | Must be improved | Not applicable | |
Does the introduction provide sufficient background and include all relevant references? | (x) | ( ) | ( ) | ( ) |
Is the research design appropriate? | (x) | ( ) | ( ) | ( ) |
Are the methods adequately described? | (x) | ( ) | ( ) | ( ) |
Are the results clearly presented? | (x) | ( ) | ( ) | ( ) |
Are the conclusions supported by the results? | (x) | ( ) | ( ) | ( ) |
Comments and Suggestions for Authors
Paper still needs reviewing by a native english speaker.
For example:
"Even if, a recent study of Ehr et al. 2017, by evaluating the transfer of ALA, EPA"
"In fact, Total fat percent varied from"
These are not the only occurrences, just examples.
The linguistic revision of whole manuscript was carried out, including the above mentioned examples.
Submission Date
21 February 2019
Date of this review
29 Mar 2019 20:41:0
